

# CMTM6 is highly expressed in lung adenocarcinoma and can be used as a biomarker of a poor diagnosis

Daqi Jia[1],[*], Li Xiong[2],[*], Honggang Xue[3] and Jidong Li[2]

[1] Department of Pathology, Kunming Medical University, Kunming, Yunan, China
[2] Department of Anesthesiology, The Third People's Hospital of Yibin, Yibin, Sichaun, China
[3] Department of Respiratory and Critical Care Medicine, General Hospital of Fuxin Mining Industry Group of Liaoning Health Industry Group, Fuxin, Liaoning, China
[*] These authors contributed equally to this work.

Corresponding author
Jidong Li, leescyb@163.com

## ABSTRACT

**Background:** CMTM6 which is chemokine-like factor (CKLF)-like Marvel transmembrane domain containing family member 6 is involved in the occurrence and progression of various tumors. However, the role of CMTM6 is still unclear in lung adenocarcinoma (LUAD).
**Methods:** Immunohistochemical, Western blotting and RT-PCR methods were used to detect the expression of CMTM6 in LUAD. Cox regression and the Kaplan-Meier method were performed to assess overall survival. Immunogenic features were evaluated according to immune cell infiltrations, immune checkpoints. The sensitivity to chemotherapy agents was estimated using the pRRophetic package.
**Results:** In LUAD, the expression of CMTM6 was obviously upregulated and was significantly associated with T stage ($p = 0.008$) and lymph node metastasis ($p = 0.018$). Multivariate Cox regression analysis demonstrated that CMTM6 was a specialty prognostic risk factor. Based on GSEA enrichment analysis, we found that high expression of CMTM6 is associated with multiple immune signaling pathways. The group with high CMTM6 expression showed a positive association with various types of tumor-infiltrating cells. Moreover, a total of 36 chemotherapeutic drugs were significantly correlated with the expression of CMTM6. Among them, two chemotherapeutic drugs had better therapeutic effects in the high CMTM6 expression group, while 34 chemotherapeutic drugs had therapeutic effects in the low CMTM6 expression group.
**Conclusion:** This study confirmed that CMTM6 is highly expressed in LUAD and is a new independent poor prognostic factor. In addition, the high expression of CMTM6 is closely related to the tumor microenvironment and immunotherapy, providing new ideas for the treatment of posterior LUAD.

## INTRODUCTION

Lung cancer remains the first and second major cause of cancer in China and the United States, respectively (*Liu et al., 2020b*; *Yang et al., 2020*). The latest data from Cancer
Statistics reported that approximately 228,820 new lung cases and bronchial cancer were diagnosed in 2020. Deaths related to lung cancer numbered 72,500 in men and 63,220 in women, for a total of 135,720 deaths (*Siegel, Miller & Jemal, 2020*). The 5-year survival rate for lung cancer patients is merely 19% for all stages combined compared (*Siegel, Miller & Jemal, 2020*). Non-small cell lung cancer is the main subtype of lung cancer, accounting for 85% of total lung cancer cases (*Xia et al., 2022*). Non-small cell lung cancer mainly includes adenocarcinoma, squamous cell carcinoma and large cell carcinoma. Of these, lung adenocarcinoma (LUAD) accounts for 50% of non-small cell lung cancers (*Chen et al., 2014*). As an important immune checkpoint, PD-1/PD-L1 has been targeted by a variety of drugs in clinical practice. For instance, nivolumab and pembrolizumab were developed to target PD-1, while atezolizumab, avelumab, and durvalumab specifically target PD-L1 (*Yang, Yu & Lu, 2020*). These drugs significantly improve the survival rates of patients with lung cancer; however, some patients are insensitive or resistant to PD-L1 inhibitors. Two studies (*Burr et al., 2017*; *Mezzadra et al., 2017*) revealed that chemokine-like factor (CKLF)-like MARVEL transmembrane domain containing family member 6 (CMTM6) plays a key role in the regulation of the PD-L1 protein. As an extensively expressed protein, CMTM6 combines with PD-L1 and is expressed on the cell surface. The downregulation of CMTM6 decreased PD-L1 without affecting MHC class I cell surface expression. Recent studies confirmed that in a variety of cancers, PD-L1 and CMTM6 are coexpressed, including gastric cancer (*Li et al., 2020*), liver cancer (*Liu et al., 2020a*), head and neck squamous cell carcinoma (*Chen et al., 2020*), and lung cancer (*Gao et al., 2019*; *Koh et al., 2019*; *Shang et al., 2020*; *Wang et al., 2020a*; *Zugazagoitia et al., 2019*). Therefore, understanding the role of CMTM6 may be beneficial to cancer patients resistant to PD-L1 inhibitors.

The relationship of CMTM6 with prognosis in lung cancer is still controversial; *Koh et al. (2019)* and *Zugazagoitia et al. (2019)* demonstrated that CMTM6 cannot be an independent risk factor for predicting lung cancer occurrence, while *Wang et al. (2020a)* reported that LUAD patients with high CMTM6 expression have a better prognosis. Recently, *Hou et al. (2020)* indicated that CMTM6 is an independent prognostic factor for NSCLC patients ($p = 0.002$). However, the sample size included in Hou's study was very small. Therefore, the role of CMTM6 in LUAD needs to be further explored. In addition, the molecular mechanism of CMTM6 is unclear in LUAD.

In the current study, we investigated the association between CMTM6 expression and clinicopathological features and explored the prognostic value of CMTM6 expression in LUAD. Then, gene set enrichment analysis (GESA) was used to predict the signaling pathways associated with high CMTM6 expression. Moreover, the association between CMTM6 and tumor-infiltrating immune cells (TIICs) was identified at the mRNA level. Our investigation provides important evidence for further study of the role of CMTM6 in LUAD.

## MATERIALS AND METHODS

### Cell culture and retrospective study

The human adenocarcinoma lung cancer cell lines H358 and A549 and the human normal lung cell line Beas-2B were purchased from The Cell Bank of Type Culture Collection of Chinese Academy of Sciences and cultured at 37 °C with 5% $CO_2$. In this study, we conducted a retrospective study. Fifty-five tumor tissues with pathologically confirmed LUAD and corresponding paired normal tissues were obtained. All paraffin-embedded tissues were collected prospectively from patients at Yibin Third People's Hospital, from September 2012 to June 2021. Fifty-five pairs of LUAD tissues were used for immunohistochemical experiments. All paraffin tissue samples are kept by our hospital, and there was thus no need to consult the patient; however, this study was approved by the Ethics Board of Yibin Third People's Hospital (2021010).

### Western blotting and RT-PCR

CMTM6 expression was detected by RT-PCR and Western blotting in A549 and H358 lung cancer cells, and Beas-2B normal lung cells were selected as the control group. Briefly, RIPA buffer with PMSF was used to lyse the cells for 15 min to extract all cell proteins. The protein solution was separated by SDS–PAGE and transferred to polyvinylidene fluoride (PVDF) membranes. The PVDF membrane was incubated with anti-CMTM6 (#D260396-0100; Sangon Biotech, Shanghai, China) at 1:1,000 at room temperature and then incubated with secondary antibody at a 1:10,000 dilution. The internal control was β -actin ( #TA-09; ZSGB-BIO, Beijing, China). According to the protocol, total cellular RNA was extracted from tumor tissues using a Relia-Prep™ RNA Cell Miniprep kit (Promega, Madison, WI, USA). The primers were as follows: CMTM6, forward 5′-G GCAACAATATCAGCAACTTC 3′ and reverse 5′-GGTCCTTAGGTGTGGTATC-3′; GAPDH, forward 5′-G GCAACAATATCAGCAACTTC-3′ and reverse 5′-GGTCCTTA GGTGTGGTATC-3′; and GAPDH, forward, 5′-ACAACAGCCTCAAGATCAT-3′; and reverse 5′-AGTCCTTCCACGATACCA-3′. Bio-Rad CFX96 Manager software was used to read the PCR amplification products. All western blotting and RT-PCR experiments were repeated at least three times.

### Immunofluorescence

The immunofluorescence method was partially based on our research group's previously published articles (*Wang et al., 2020b*). Tumor cells grown on glass slides were fixed with 4% paraformaldehyde, permeabilized with PBS containing 0.2% Triton X-100 (Sigma–Aldrich, Darmstadt, Germany) and washed with PBS containing 0.02% Tween-20 (PBST) three-times. After incubation for 45 min in 10% BSA diluted in PBST, the cells were incubated with CMTM6 antibody (#D260396-0100; Sangon Biotech, Shanghai, China) at 4 °C overnight in a humidified chamber. After washing with PBST three times, the cells were incubated with goat anti-rabbit IgG (diluted 1:150 with 0.01 mol PBS) conjugated with FITC (ZF-0311; Zhong Shan Jin Qiao, Beijing, China) at 37 °C for 40 min in a humidified chamber in the dark. Nuclei were stained with DAPI (C1005; Beyotime,

Shanghai, China) at 25 °C for approximately 15–20 min. The images used for analysis were obtained with fluorescence microscopy (Olympus BX51, Tokyo, Japan).

## Immunohistochemistry

Human primary lung adenocarcinoma tissues were fixed in formalin and embedded in paraffin. Then, combined with HE-stained sections, the corresponding cancer tissue or adjacent tissue was embedded in paraffin to generate a tissue microarray. Five-micron-thick tissue sections were incubated with CMTM6 antibodies (#D260396-0100; Sangon Biotech, Shanghai, China) (diluted 1:50 in 0.01 mol/L TBST with 1% BSA) overnight at 4 °C and then washed in 0.01 mol/L PBS. The sections were then exposed to a monoclonal antibody conjugated with horseradish peroxidase (ZSGB-BIO, Beijing, China) for 30 min and washed with PBS. Finally, the slides were stained with 3,3-diaminobenzidine and hematoxylin.

## Data from the TCGA cohort

The RNA-seq data and the relevant clinical information of LUAD patients were acquired from The Cancer Genome Atlas database (TCGA) (https://portal.gdc.cancer.gov/). Patients were divided into high- and low-expression groups according to the median value of each sample's CMTM6 mRNA expression. For clinical analysis, the follow-up time of all cases varied from 90 days to 10 years, and cases with uncertain cancer type and unknown age were excluded.

## Expression and clinical association analyses for CMTM6 in LUAD

In our study, based on TCGA data, we first analyzed the expression difference between lung tumor and normal lung tissues using the Wilcoxon signed-rank test. Kaplan–Meier curves were then used to estimate the overall survival (OS) of CMTM6 expression for lung cancer patients according to its l status. Finally, univariate and multivariate Cox analyses were selected to compare the effects of CMTM6 expression *vs* other clinical characteristics on survival. The connection between clinical factors and CMTM6 was analyzed using logistic regression. The online websites Kaplan–Meier Plotter (https://kmplot.com/analysis/) and PrognoScan (http://dna00.bio.kyutech.ac.jp/PrognoScan/index.html) were also selected to analyze the prognostic role of CMTM6 in lung adenocarcinoma.

## Gene set enrichment analysis (GSEA)

GESA can build a list of all genes related to the expression of one gene (*Subramanian et al., 2005*) The h.all.v7.5.symbols.gmt (Hallmarks) gene set database was selected in this study. GSEA was used to elaborate the functional differences of the gene between the low- and high-CMTM6 groups, with CMTM6 expression being considered a phenotype. The arrangements of gene sets were arranged out by 1,000 times per every analysis. The FDR $q$ value, the normalized enrichment score (NES) and the nominal $p$ value were established to classify each phenotypic enrichment signaling pathway.

## Correlation analysis between CMTM6 expression and tumor-infiltrating immune cells in LUAD

CMTM6 is a significant immune-related gene (*Burr et al., 2017*; *Mezzadra et al., 2017*). To determine the related immune population abundance in high- and low-CMTM6 lung cancer patients, CIBERSORT (*Newman et al., 2015*) with the standard LM22 signature gene file and 1,000 arrays were implemented. The CIBERSORT deconvolution algorithm is a relatively exact and robust method that estimates 22 types of immune cells (dendritic cells, natural killer cells, B cells, T cells, plasma cells, mast cells and macrophages) through microarray gene expression data. The results with a CIBERSORT $p$ value < 0.05 were selected and enrolled for further analysis.

## Correlation analysis between CMTM6 expression and immune checkpoint-related genes in LUAD

A total of 91 immune checkpoint-related genes (ICGs) were identified through a literature search (Table S1). Based on the RNA-seq data of LUAD in TCGA, we analyzed the correlation between CMTM6 and ICGs to explore immune genes related to CMTM6 expression. The R package used in this section included limma, reshape2, ggplot2, ggpubr, and corrplot. The correlation threshold was set to a correlation coefficient $\geq 0.2$ and $p$ value < 0.05. We further verified ICGs with a high correlation with CMTM6 through the GEPIA2 website (http://gepia2.cancer-pku.cn/#index).

## Analysis of immunogenicity and chemosensitivity

The immunophenoscore (IPS) data of TCGA-LUAD tumor samples were acquired from The Cancer Immunome Atlas (TCIA, https://tcia.at/home) database. The Wilcoxon test was used to test the differences between the low- and high-risk groups to further evaluate the sensitivity of LUAD samples to PD1/PDL1 and CTLA4 antibodies. The therapeutic responses to known chemotherapy agents were estimated using the pRRophetic package (*Geeleher, Cox & Huang, 2014*).

## Statistical analysis

Data with a normal distribution were compared by t tests, and data with a nonnormal distribution were compared by nonparametric tests. CMTM6 expression was evaluated by the percentage of positive cells and histological score (HSCORE) (*Budwit-Novotny et al., 1986*). The positive cell rate was determined *via* IHC, and further analysis was conducted with both a $\chi^2$ test and Fisher's exact test. R software 4.0 (Bell Laboratories, Lucent Technologies, Co., Ltd., Morristown, NJ, USA) and GraphPad Prism software 9.0 (GraphPad Software Co., Ltd., San Diego, CA, USA) were used to complete all statistical analyses.
**Table 1 Expression of CMTM6 protein in LUAD and adjacent tissues.**

|  | Cases | Expression of CMTM6 | | | $\chi^2$ | p value |
|---|---|---|---|---|---|---|
|  |  | Negative | Positive | Positive rate (%) |  |  |
| Adjacent | 55 | 40 | 15 | 27.27 | 17.65 | <0.0001 |
| Tumor | 55 | 18 | 37 | 67.27 |  |  |

# RESULTS

## CMTM6 expression in LUAD was significantly higher than that in adjacent tissues

The tissue microarray and immunohistochemical assays were performed on 55 paired cancer and adjacent tissues, and three independent pathologists interpreted the negative and positive results according to the immunohistochemical results and determined the H-score for each case. Among 55 cases of LUAD, CMTM6 expression was positive in 37 cases (67.27%) and negative in 18 cases (32.73%). Among the 55 cases of corresponding adjacent tissues, CMTM6 expression was positive in 15 cases (27.27%) and negative in 40 cases (72.73%) (Table 1). Through the chi square test, the results showed a significant difference in CMTM6 expression between LUAD and adjacent tissues (chi-square: 17.65, df: 1, $p < 0.0001$) (Table 1 and Fig. 1A). Through the H-score of the immunohistochemical results of the cases, the results showed that the cancer tissue was significantly higher than the adjacent tissues. The H-score showed that the distribution of cancer tissue was mainly concentrated above 100 points, while the distribution of adjacent tissues was mainly concentrated below 100 points (Fig. 1B). The immunohistochemical results of CMTM6 in paracancerous tissues, low-expression lung adenocarcinoma and high-expression lung adenocarcinoma are shown in Fig. 1C. To further explore the expression of CMTM6 in lung adenocarcinoma, we extracted the LUAD mRNA expression in TCGA database. The results showed that CMTM6 expression was also significantly higher in LUAD than in adjacent tissues (Figs. 2A and 2B). Western blot and qPCR results showed that CMTM6 expression was significantly higher in the LUAD cell lines A549 and H358 than in BEAS-2B normal lung epithelial cells (Figs. 2C and 2D). Immunofluorescence experiments were carried out on A549 and H358 cells. The results showed that CMTM6 was significantly overexpressed and localized in the cell membrane and pericellular system (Figs. 2E and 2F).

## Correlation between CMTM6 expression and clinicopathological features of lung adenocarcinoma and its prognostic value in LUAD

Based on the clinical samples we collected, we analyzed the correlation between CMTM6 expression and patient age, sex, tumor differentiation, T stage and lymph node metastasis in the LUAD group. The results showed that CMTM6 expression was significantly correlated with tumor T stage (chi-square: 9.729, df: 2, $p = 0.008$) and lymph node metastasis (chi-square: 5.593, df: 1, $p = 0.018$) and had no clinical correlation with other clinical indicators (Table 2). The prognosis analysis based on the clinical data of TCGA

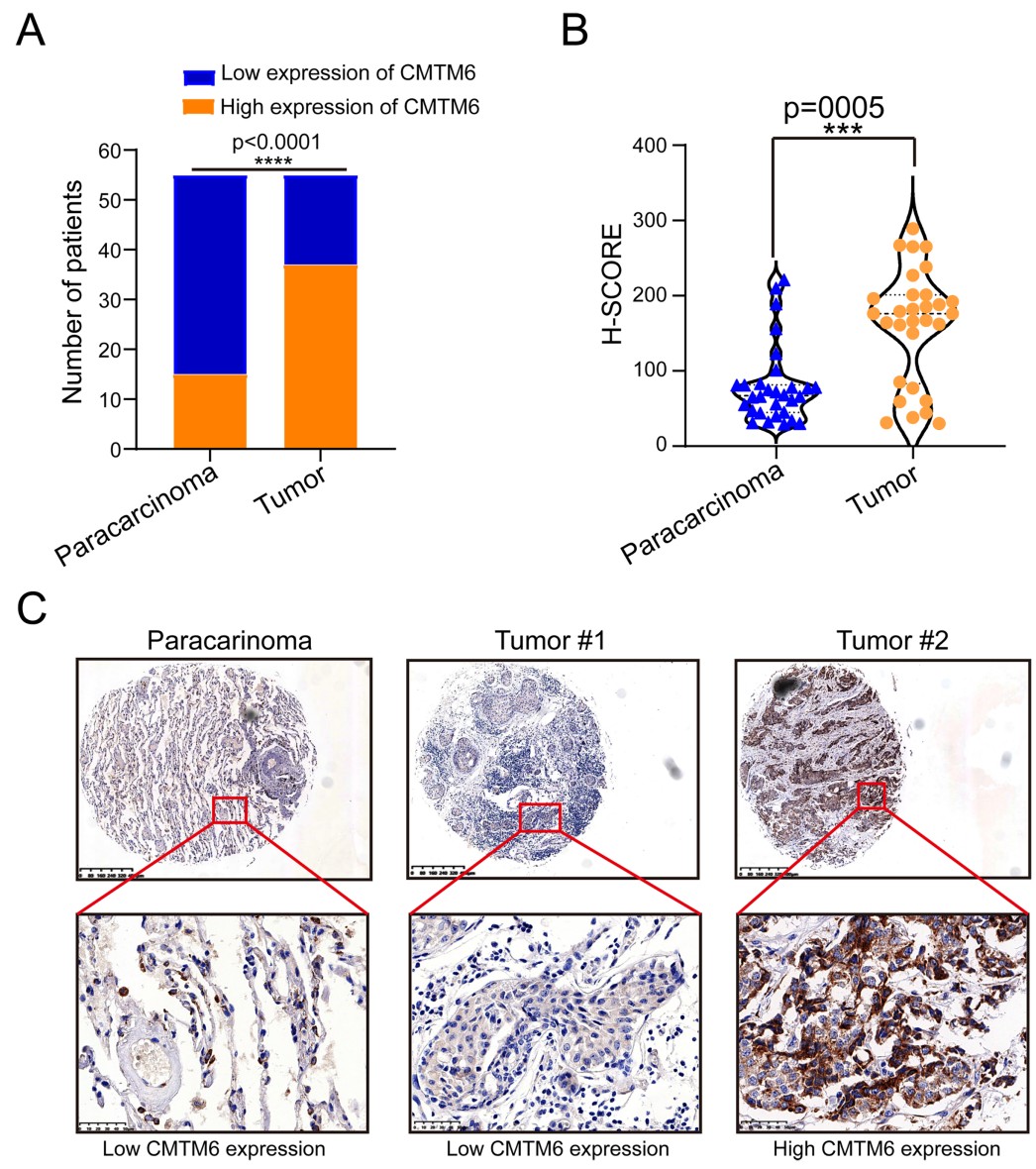

**Figure 1 CMTM6 is upregulated in LUAD tissues based on IHC.** (A) Statistical immunohistochemical results showed that the number of positive cases of CMTM6 in lung adenocarcinoma was significantly higher than that in adjacent cases. (B) H-SCORE was evaluated according to immunohistochemical results and a violin plot was drawn. The results showed that the HSCORE in the tumor group was significantly higher than that in the paracancer group. (C) The expression of CMTM6 in paracancer tissue tumor tissue in the tissue microarray showed that CMTM6 was mainly located in the cell membrane and cytoplasm. The statistical method chosen in this part is the paired T-test.

LUAD showed that patients with high CMTM6 expression had a short survival time (Fig. 3A). Prognosis analysis based on the Kaplan–Meier Plot online website showed that the prognosis of patients with high CMTM6 expression LUAD was poor, and the HR and 95% CI were 1.45 [1.13–1.86] (Fig. 3B). PrognoScan website prognosis analysis of overall survival results showed that patients with high CMTM6 expression had a poor prognosis (Fig. 3C). PrognoScan website prognosis analysis relapse-free survival results showed that

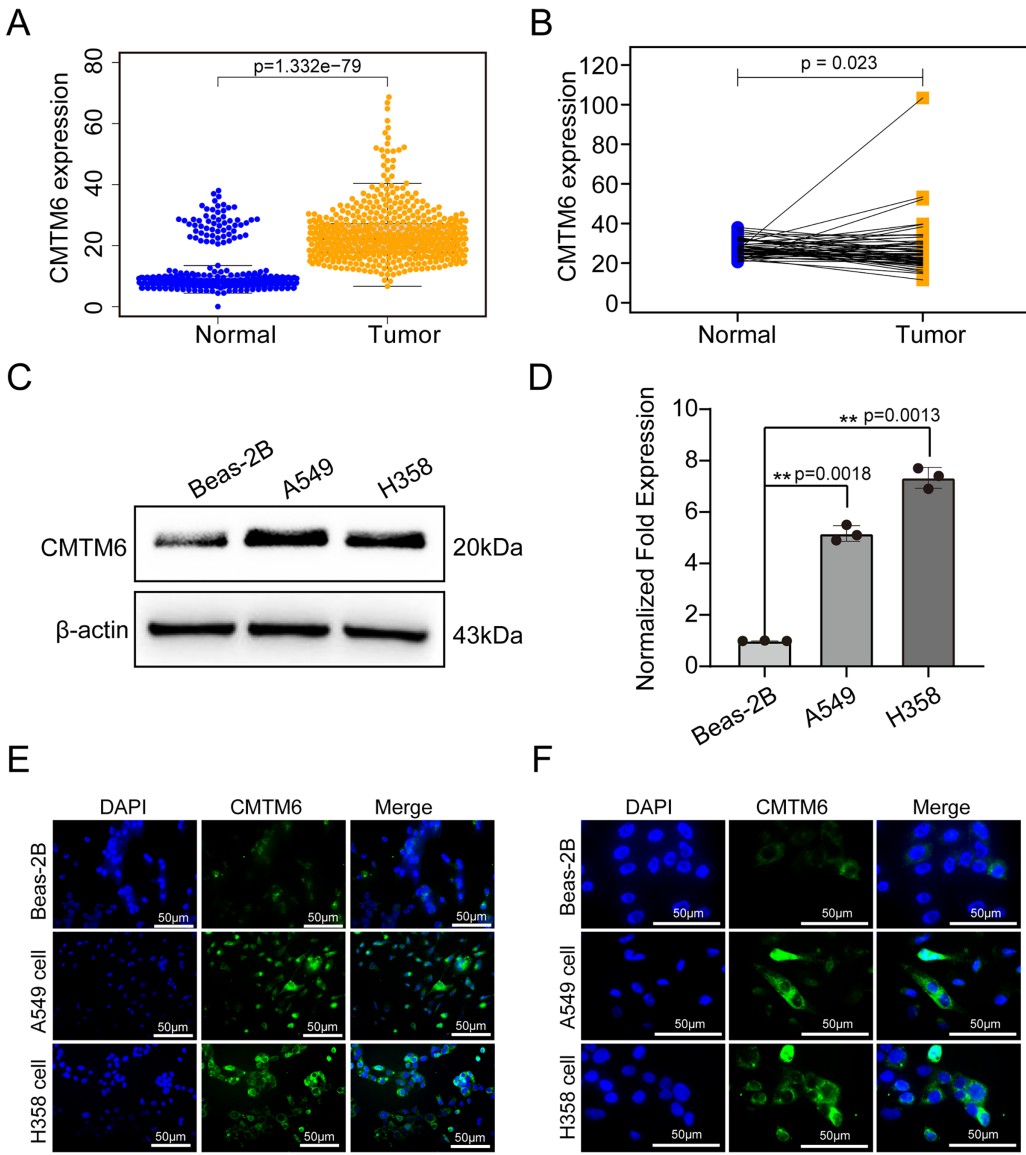

**Figure 2 CMTM6 was significantly upregulated in LUAD based on TCGA combined with the GTEx database and cell experiments.** (A) The scatter plot from TCGA combined with the GTEx database depicted the differential CMTM6 expression between normal tissues and tumor tissues. (B) Based on the same sample number, CMTM6 expression in tumor and normal tissues was compared. (C) Western blotting was used to detect the protein expression of CMTM6 in LUAD cells and normal lung epithelial cells. (D) RT–PCR was used to detect the mRNA expression of CMTM6 in LUAD cells and normal lung epithelial cells. An immunofluorescence assay was used to detect the expression and localization of CMTM6 in Beas-2B, A549 and H358 cells, and the cells were observed under a 40× (E) fluorescence microscope and 100× (F) fluorescence microscope.

patients with high CMTM6 expression also had a poor prognosis (Fig. 3D). Univariate Cox analysis indicated that high CMTM6 expression was significantly positively correlated with poor prognosis. Multivariate analysis based on TCGA clinical data further confirmed that patients with high CMTM6 expression could be used as an independent prognostic factor (Figs. 3E and 3F).

**Table 2 Relationship between the expression level of CMTM6 and clinicopathological parameters of LUAD.**

| Clinicopathological characteristics | Cases | CMTM6 expression | | $\chi^2$ | p value |
|---|---|---|---|---|---|
| | | Low | High | | |
| Gender | | | | | |
| Male | 33 | 10 | 23 | 0.22 | 0.639 |
| Female | 22 | 8 | 14 | | |
| Years | | | | | |
| <60 | 32 | 11 | 21 | 0.094 | 0.759 |
| >=60 | 23 | 7 | 16 | | |
| Differentiation | | | | | |
| High | 21 | 3 | 18 | 5.454 | 0.065 |
| Medium | 19 | 9 | 10 | | |
| Low | 15 | 6 | 9 | | |
| T stage | | | | | |
| T1 | 16 | 10 | 6 | 9.729 | 0.008 |
| T2 | 14 | 4 | 10 | | |
| T3 | 25 | 4 | 21 | | |
| Lymph node metastasis | | | | | |
| N0 | 17 | 9 | 8 | 5.593 | 0.018 |
| N1 | 38 | 8 | 30 | | |

## Gene set enrichment analysis

In LUAD, CMTM6 expression was analyzed using GSEA. Overexpression of CMTM6 and tumorigenesis mainly included angiogenesis (NES = 1.66; NOM p value = 0.041; FDR q-value = 0.055), IL6 JAK STAT3 signaling (NES = 1.74; NOM p value = 0.033; FDR q-value = 0.040), KRAS signaling (NES = 2.10; NOM p value = 0.002; FDR q-value = 0.003), PI3K AKT MTOR signaling (NES = 1.87; NOM p value = 0.002; FDR q-value = 0.022), and TGF beta signaling (NES = 2.19; NOM p value < 0.0001; FDR q-value = 0.003) (Fig. 4A). The nontumor-related signaling pathways involved in the high expression of CMTM6 mainly included allograft rejection (NES = 1.77; NOM p value = 0.034; FDR q-value = 0.035), androgen response (NES = 2.05; NOM p value < 0.0001; FDR q-value = 0.015), complement (NES = 1.89; NOM p value = 0.004; FDR q-value = 0.019), heme metabolism (NES = 1.73; NOM p value < 0.0001; FDR q-value = 0.038), inflammatory response (NES = 2.11; NOM p value = 0.004; FDR q-value = 0.005), protein secretion (NES = 2.27; NOM p value < 0.0001; FDR q-value = 0.002) and UV response (NES = 1.91; NOM p value = 0.006; FDR q-value = 0.019) (Fig. 4B).

## Increased CMTM6 expression activates immune cell infiltration

The relationship between immune cell levels and CMTM6 expression in LUAD was examined *via* the TIMER database. The levels of CD8+ T cells, CD4+ T cells, macrophages, neutrophils and dendritic cells were significantly positively correlated with CMTM6

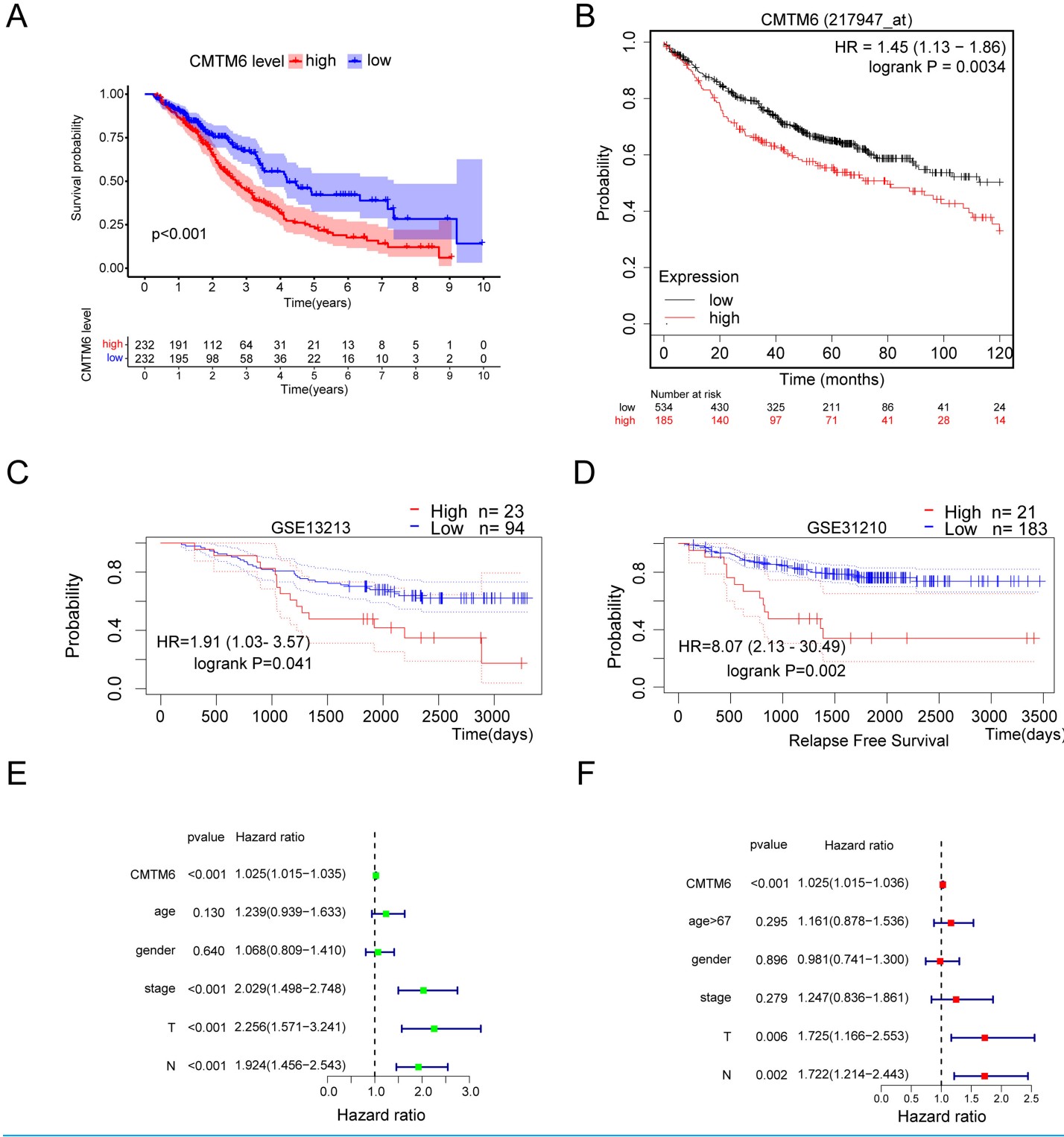

**Figure 3 Lung adenocarcinoma with high CMTM6 expression has a poor prognosis.** (A) Overall survival was analyzed based on clinical data of LUAD and CMTM6 expression in TCGA. (B) The Kaplan-meier Plotter online website was used to analyze the prognosis of CMTM6 in lung adenocarcinoma. (C and D) The PrognoScan online website was also selected to analyze the prognostic role of CMTM6 in lung adenocarcinoma. (E) The prognosis of CMTM6 was analyzed by univariate Cox regression. (F) Multivariate Cox regression analysis of the prognosis of CMTM6.

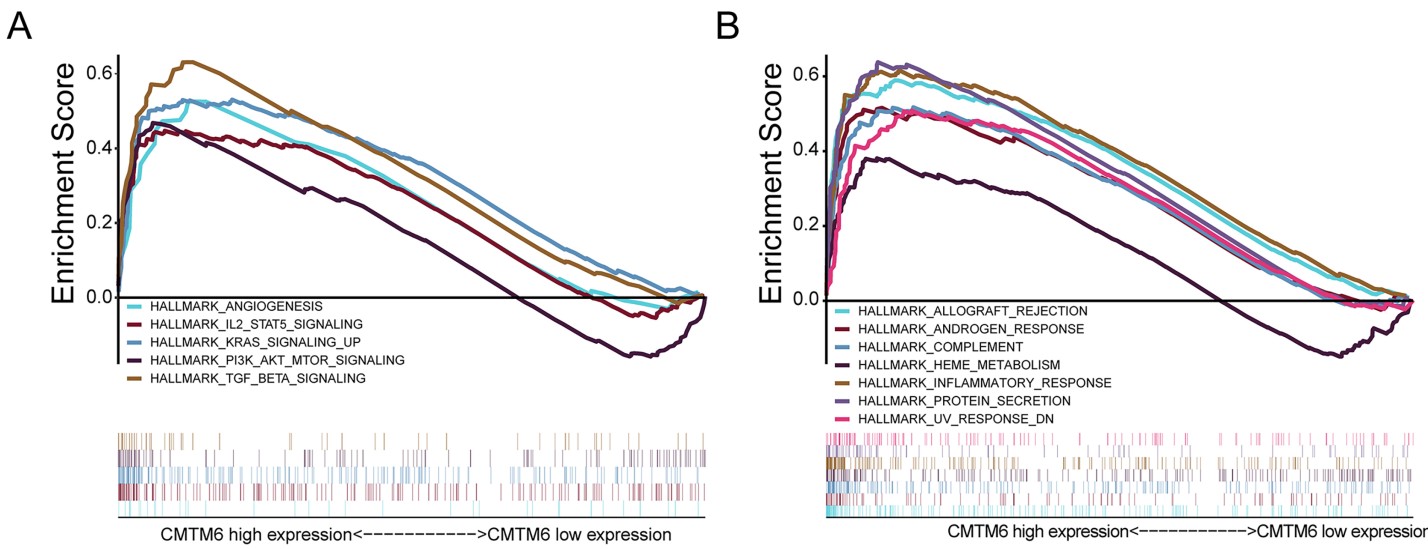

**Figure 4 Analysis of signaling pathways by using GSEA.** (A) GSEA plots showing the NES for HALLMARK, which is closely related to the overexpression of CMTM6 in lung adenocarcinoma and involves multiple-tumor related signaling pathways, including angiogenesis, IL6 JAK STAT3 signaling, KRAS signaling, PI3K AKT MTOR signaling, and TGF beta signaling. (B) Nontumor signaling pathways associated with overexpression of CMTM6 in lung adenocarcinoma include allograft rejection, androgen response, complement, heme metabolism, inflammatory response, protein secretion and UV response.

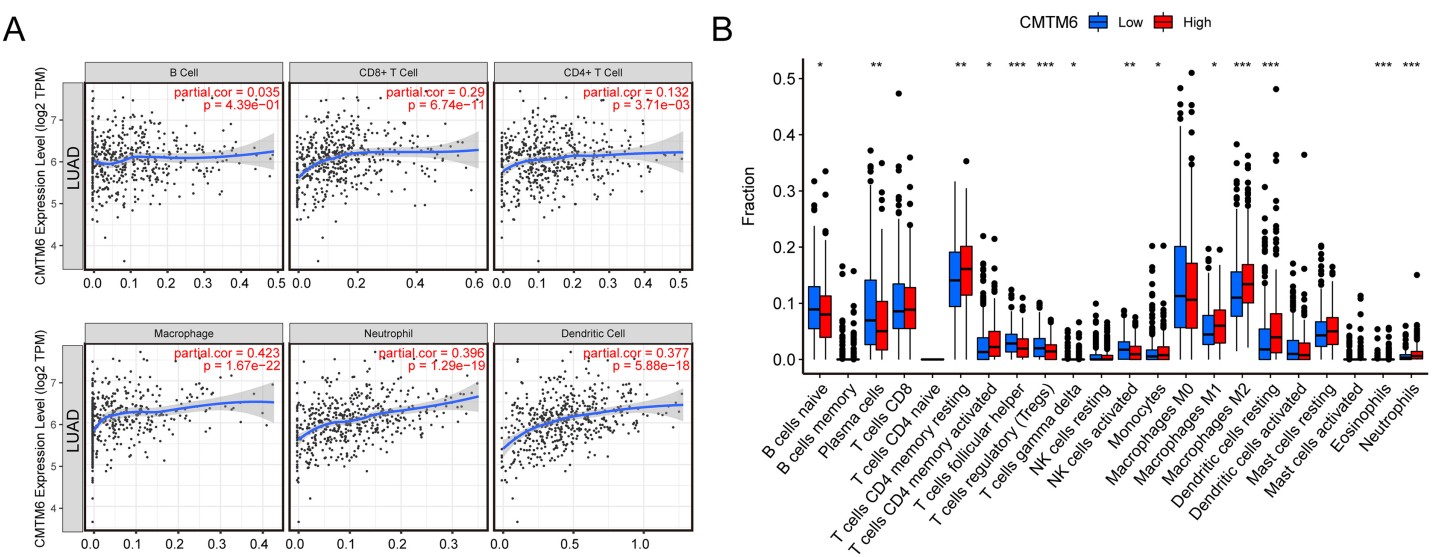

**Figure 5 Increased CMTM6 expression increased immune cell infiltration.** (A) The infiltration levels of B cells, CD8+ T cells, CD4+ T cells, macrophages, neutrophils and dendritic cells were significantly negatively correlated with CMTM6 expression. (B) The enrichment scores of 22 types of immune cells across the expression of CMTM6.

expression. However, there was no significant correlation between B cells and CMTM6 expression (Fig. 5A). Moreover, when analyzing whether there was a difference between CMTM6 expression and immune cells, the results showed a significant difference between CMTM6 expression and naïve B cells, plasma cells, resting memory CD4 T cells, activated memory CD4 T cells, follicular helper T cells, regulatory T cells (Tregs), activated gamma

delta T cells, activated NK cells, monocytes, M1 macrophages, M2 macrophages, resting dendritic cells, eosinophils and neutrophils (Fig. 5B).

## Correlation analysis of CMTM6 and ICGs

Based on the RNA-seq data of LUAD in TCGA, we analyzed the correlation between CMTM6 and ICG expression. The results showed that the correlation between CMTM6 and 17 ICGs was greater than or equal to 0.2 and negatively correlated with 3 ICGs (Figs. 6A and 6B). Based on the GEPIA2 website, the correlation between CMTM6 and ICGs was further verified. The results showed that the correlation coefficients between CMTM6 and PDCD1LG2, HAVCR2, CD200R1, TNFRSF4, TNFRSF18 and TNFRSF25 were 0.32, 0.38, 0.35, −0.12, −0.15 and −0.12, respectively (Figs. 6C–6H).

## CMTM6 expression correlates with immunogenicity and chemosensitivity of LUAD

The immunogenicity of CMTM6 was analyzed by immunophenoscore analysis. The ips_ctla4_pos_pd1_pos and ips_ctla4_neg_pd1_pos scores were not significantly different between the high- and low-CMTM6 expression groups (Figs. 7A and 7B). However, ips_ctla4_pos_pd1_neg and ips_ctla4_neg_pd1_neg scores were higher in the low-CMTM6-expression groups, indicating that immunotherapy is more effective in the low-CMTM6-expression group in LUAD (Figs. 7C and 7D). Furthermore, we observed that high CMTM6 expression had a strong positive correlation with TME scores (Fig. 7E). According to the pRRophetic package, further analysis was conducted to evaluate the correlations between CMTM6 expression and chemosensitivity of LUAD. Nine lung cancer related-chemotherapy drugs, rapamycin, pazopanib, paclitaxel, dasatinib, TGX221, saracatinib, sunitinib, ruxolitinib, and erlotinib were evaluated in this study. howed that low CMTM6 expression was associated with increased sensitivity to rapamycin, pazopanib, paclitaxel, dasatinib, TGX221, saracatinib, sunitinib, and ruxolitinib. Only the chemotherapeutic drug erlotinib showed better therapeutic effects in the high-CMTM6-expression group (Fig. 8).

## DISCUSSION

In China and the United States, lung cancer is the leading cause of cancer-related death; its occurrence is complicated and involves multiple signaling pathways, abundant genetic expression, and mutations (*Siegel, Miller & Jemal, 2020*). Although numerous prognostic markers have been identified, the prognosis of patients with lung cancer, especially advanced patients, is still poor. Recently, inhibitors of immune checkpoints (such as PD-1/PD-L1) have exhibited promising therapeutic outcomes and have been approved for a variety of cancer treatments (*Darvin et al., 2018*). Compared with chemotherapy drugs, several immune checkpoint inhibitors can significantly prolong the survival time of lung cancer patients. For example, in LUSC patients, nivolumab, a PD-1 inhibitor, showed significantly better efficacy and safety than docetaxel (objective response rate (ORR): 20% *vs* 9%; *p* = 0.008) (*Brahmer et al., 2015*) and advanced NSCLC patients (ORR: 19% *vs* 12%; *p* = 0.02) (*Borghaei et al., 2015*). Furthermore, durvalumab, another PD-L1 inhibitor,

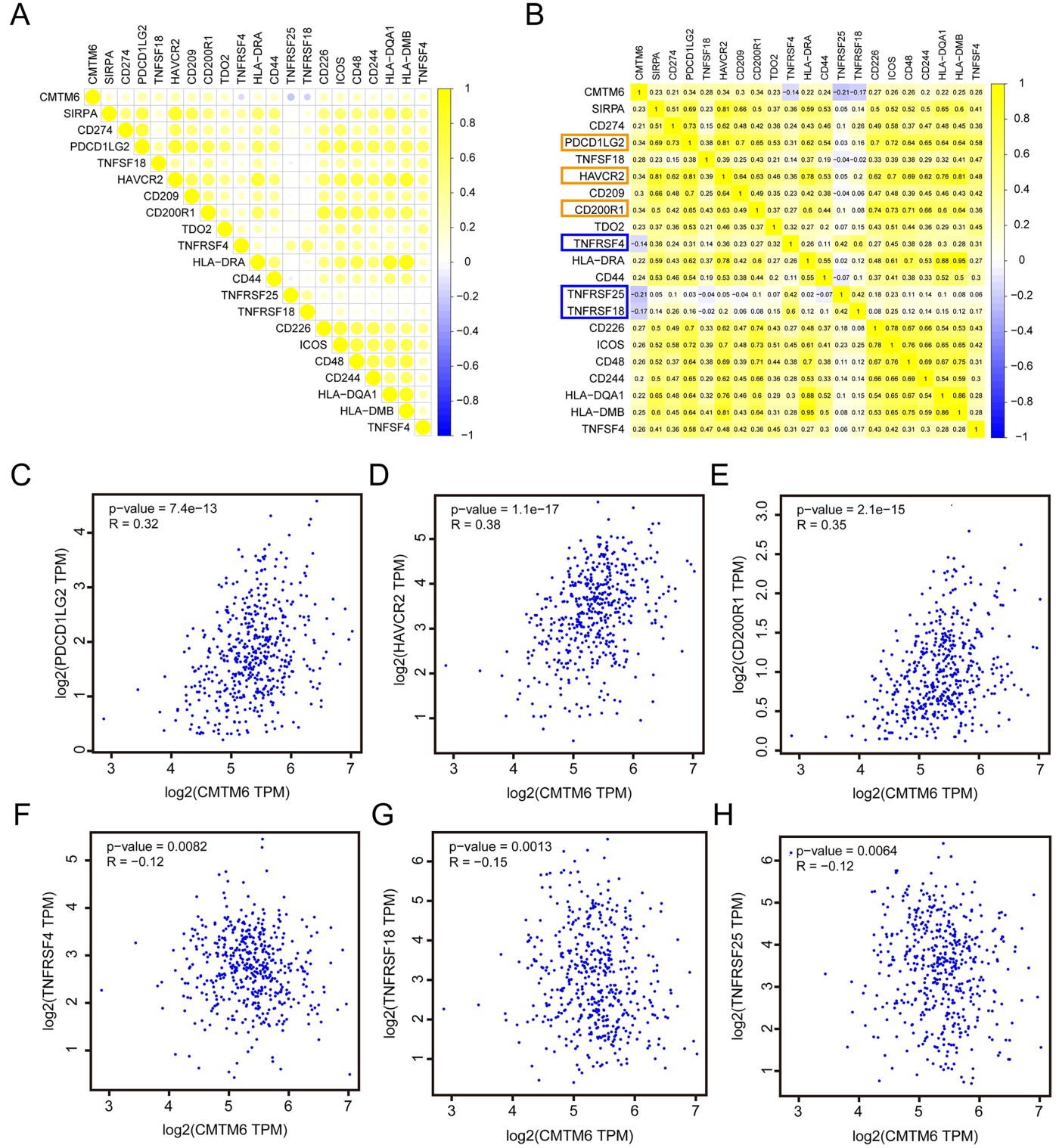

**Figure 6 Correlation analysis of CMTM6 and immune checkpoint-related gene expression.** (A and B) TCGA LUAD database was used to analyze the correlation between CMTM6 and the expression of 91 immune checkpoint-related genes, and the correlation coefficient threshold was set at 0.2. (C–E) Based on GEPIA2 website analysis, CMTM6 was significantly positively correlated with PDCD1LG2, HAVCR2 and CD200R1 expression. (F–H) Based on GEPIA2 website analysis, CMTM6 was significantly negatively correlated with TNFRSF4, TNFRSF18 and TNFRSF25 expression.

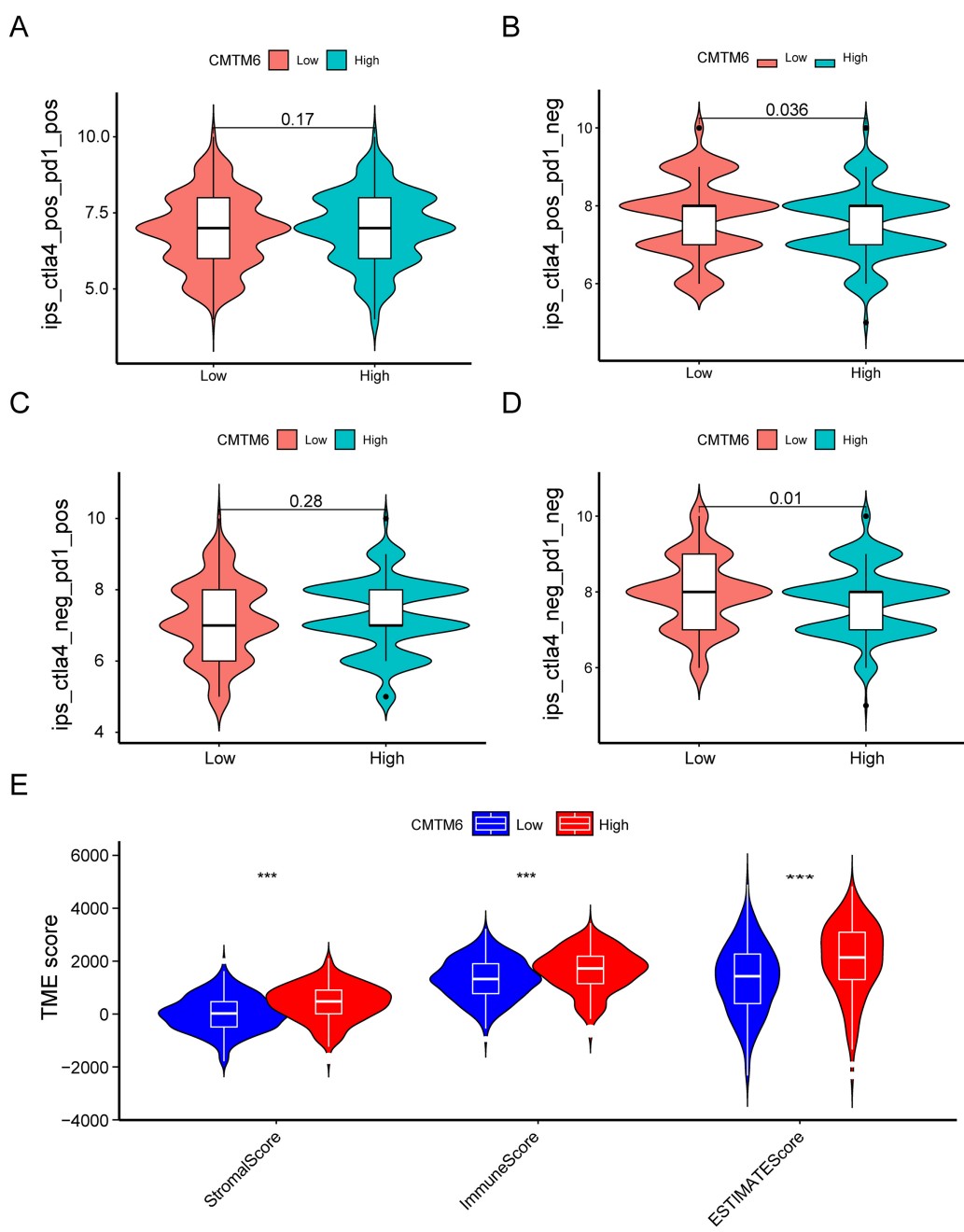

**Figure 7 Immune landscape between the high- and low-CMTM6 expression groups.** (A–D) Differences in immunogenicity between the high- and low-CMTM6 expression groups. (E) The relationship between CMTM6 expression and the TME score.

can significantly improve the 2-year survival rates in the durvalumab use group of NSCLC compared to the placebo group (*Antonia et al., 2018*). Although PD-L1 inhibitors have achieved good results in lung cancer treatment, a large number of patients with advanced lung cancer are still resistant to these inhibitors. CMTM6 is a novel target regulating the PD-L1 protein, although it was first reported in 2003 (*Darvin et al., 2018*; *Li et al., 2020*).

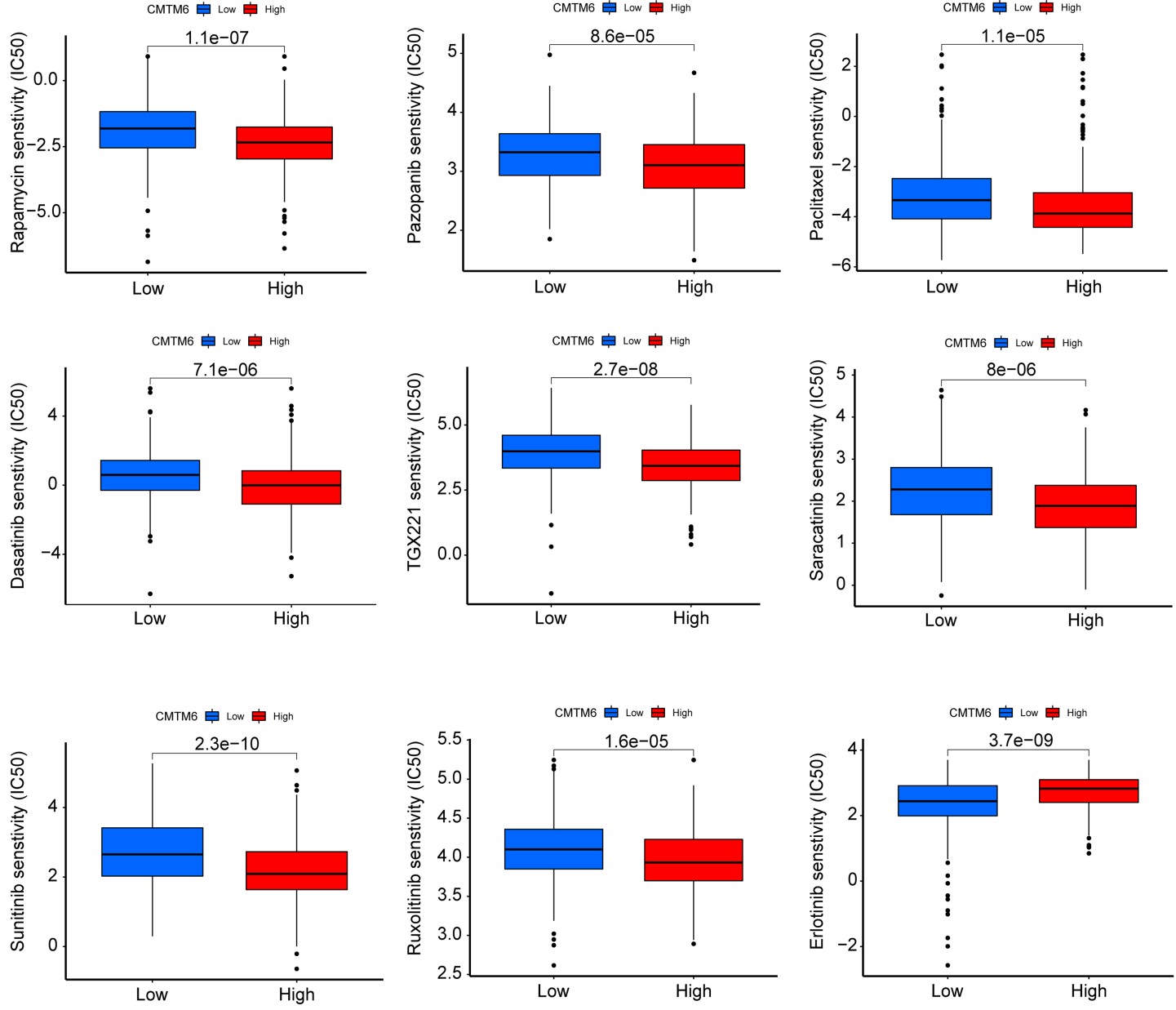

**Figure 8 CMTM6 correlates with the chemosensitivity of LUAD.** IC50 values of rapamycin, pazopanib, paclitaxel, dasatinib, TGX221, saracatinib, sunitinib, ruxolitinib, and erlotinib in high- and low-expression patients with LUAD.

Recent studies have suggested that CMTM6 is coexpressed with PD-L1 in a variety of cancers and has been proven to be an independent prognostic factor of gastric cancer and liver cancer (*Guan et al., 2018*; *Li et al., 2020*; *Liu et al., 2020a*; *Zhu et al., 2019*). Understanding the role of CMTM6 may be beneficial to cancer patients resistant to PD-L1 inhibitors.

In this study, we found a significant difference in CMTM6 expression between lung adenocarcinoma and adjacent tissues. The H-score of the immunohistochemical results showed that CMTM6 expression in cancer tissue was significantly higher than that in

adjacent tissues. TCGA combined with the GTEx database further confirmed the high CMTM6 expression in lung adenocarcinoma and low expression in paracancerous tissues. Cell experiments further confirmed our results that CMTM6 expression was significantly higher in LUAD cells than in normal lung epithelial cells. Our results are different from those of *Wang et al. (2020a)*, who reported that CMTM6 is expressed at low levels in lung adenocarcinoma. We noted that their normal samples were all based on the TCGA database. Our study enriched the number of normal control groups by including GTEx normal samples. Prognosis analysis based on the Kaplan–Meier Plot online website showed that the prognosis of patients with high CMTM6 expression in LUAD was poor, and the HR and 95% CI were 1.45 [1.13–1.86]. Multivariate analysis based on TCGA clinical data further confirmed that patients with high CMTM6 expression could be used as an independent prognostic factor. However, the role and prognostic value of CMTM6 in different tumors are different, and it is likely to be a multiple effect factor. *Yugawa et al. (2021)* reported that CMTM6 stabilizes PD-L1 expression and is a new prognostic impact factor in hepatocellular carcinoma. *Zhang, Zhao & Wang (2021)* indicated that high CMTM6 expression correlated with a poor prognosis in gastric cancer patients. The role of CMTM6 in colorectal cancer is opposite to that in liver cancer and gastric cancer. *Peng et al. (2021)* found that CMTM6 is associated with an active immune microenvironment and a favorable prognosis in colorectal cancer.

In view of the complex role of CMTM6 in tumors, it is particularly important to clarify the signaling pathways by which CMTM6 may participate in tumorigenesis. We used GSEA to explore which signaling pathway may be involved in the high expression of CMTM6 in patients. We first found that high CMTM6 expression was associated with tumorigenesis, mainly including angiogenesis, IL6 JAK STAT3 signaling, and KRAS signaling. These three signaling pathways are closely related to tumorigenesis. Whether CMTM6 participates in these three signaling pathways in other tumors has not yet been reported. We also found that overexpression of CMTM6 was involved in TGF beta signaling and PI3K AKT MTOR signaling. *Chen et al. (2020)* showed that deletion of the CMTM6 gene inhibited stem cell-like properties, TGF-β-induced EMT and proliferation of head and neck squamous cell carcinoma cells. The nontumor-related signaling pathways involved in high CMTM6 expression mainly include allograft rejection, androgen response, complement, heme metabolism, protein secretion and UV response. The correlation between the above nontumor-related signaling pathways and CMTM6 that we found has rarely been reported in other tumors. We also found that high CMTM6 expression is involved in the inflammatory response signaling pathway, which is consistent with the reports of *Guan et al. (2018)*, *Shang et al. (2020)*, *Wang et al. (2020a)* and *Wu et al. (2021)*.

CMTM6 is an important immune-related gene, and we speculate that it may have a regulatory effect on the tumor microenvironment. By analyzing the correlation between CMTM6 and immune cell infiltration, we found that CMTM6 overexpression positively affected CD8+ T cells, CD4+ T cells, macrophages, neutrophils and dendritic cells. In fact, *Mezzadra et al. (2017)* demonstrated that CMTM6 could enhance the ability of PD-L1-expressing tumor cells to suppress CD8 T cells. The interaction of CMTM6 with other

tumor-infiltrating cells has not been reported. Combined with our prediction results, we speculated that CMTM6 may participate in a variety of tumor signal-related pathways by affecting the activity of tumor-infiltrating cells. The correlation between CMTM6 and macrophages has been reported in recent years. *Pang et al. (2021)* reported that OSCC cell-secreted exosomal CMTM6 induces M2-like macrophage polarization to promote malignant progression. Direct studies on the relationship between CMTM6 and neutrophils and dendritic cells have rarely been reported. Based on the RNA-seq data of LUAD in TCGA, we analyzed the correlation between CMTM6 and ICG expression. The results showed that the correlation between CMTM6 and 17 ICGs was greater than or equal to 0.2 and negatively correlated with 3 ICGs. We found that CMTM6 was significantly and positively correlated with PD-L1 expression in lung adenocarcinoma, which was consistent with the CMTM6 and PD-L1 correlation reported in multiple tumors, including breast cancer, oral squamous cell carcinoma, hepatocellular carcinoma and gastric cancer (*Li et al., 2020*; *Liu et al., 2021*; *Tian et al., 2021*; *Zhang et al., 2021*). In addition, we found that the correlation coefficient between CMTM6 and the other three immune checkpoints (PD-L2, HAVCR2 and CD200R1) in lung adenocarcinoma was higher than that between CMTM6 and PD-L1. However, in lung adenocarcinoma, there is no report on the relationship between CMTM6 and PD-L2, HAVCR2 and CD200R1. Whether CMTM6 can affect the function of the above three immune checkpoints in lung adenocarcinoma needs further study.

The immunogenicity of CMTM6 was analyzed by immunophenoscore analysis. ips_ctla4_pos_pd1_neg and ips_ctla4_neg_pd1_neg scores were higher in the low CMTM6 expression groups, indicating that immunotherapy is more effective in the low CMTM6 expression group in LUAD. Furthermore, we observed that high CMTM6 expression had a strong positive correlation with TME scores. According to the pRRophetic package, further analysis was conducted to evaluate the correlations between CMTM6 expression and nine types of lung cancer-related chemotherapy drugs. We found that LUAD patients with high CMTM6 expression are not sensitive to eight different of chemotherapy drugs. That is, when patients with lung adenocarcinoma have high CMTM6 expression, the use of these chemotherapy drugs may not be effective, however, this findings still need to be verified by a large number of clinical cases.

There are several limitations in our study. Smoking is an important factor that stimulates the occurrence and prognosis of LUAD (*Molina et al., 2008*). Regrettably, this TCGA dataset does not contain clear smoking information. Moreover, large cell carcinoma and small cell lung cancer were not included in this study due to the small sample size. Further study is needed to clarify the correlation between the above factors and CMTM6 in LUAD.

## CONCLUSIONS

In conclusion, we confirmed that CMTM6 is significantly upregulated in LUAD tissues and that CMTM6 indicates a poor prognosis and can serve as an independent prognostic marker in LUAD. CMTM6 is involved in multiple tumor-related signaling pathways and is

closely related to various tumor-infiltrating cells. Our investigation provides significant evidence for further study of the role of CMTM6 in LUAD.

## ACKNOWLEDGEMENTS

We thank Ting-Fang Yang for editing this manuscript.

### Funding

The authors received no funding for this work.

### Competing Interests

The authors declare that they have no competing interests, Honggang Xue is employed by Liaoning Health Industry Group.

### Author Contributions

- Daqi Jia conceived and designed the experiments, performed the experiments, analyzed the data, prepared figures and/or tables, authored or reviewed drafts of the article, and approved the final draft.
- Li Xiong performed the experiments, analyzed the data, authored or reviewed drafts of the article, and approved the final draft.
- Honggang Xue analyzed the data, prepared figures and/or tables, and approved the final draft.
- Jidong Li conceived and designed the experiments, authored or reviewed drafts of the article, and approved the final draft.

### Human Ethics

The following information was supplied relating to ethical approvals (*i.e.*, approving body and any reference numbers):

This study received the approval of the Ethics Board of Yibin Third People's Hospital (2021010).

### Microarray Data Deposition

The following information was supplied regarding the deposition of microarray data:

NONE

### Data Availability

The raw measurements are available in the Supplemental Files.

### Supplemental Information

Supplemental information for this article can be found online at http://dx.doi.org/10.7717/peerj.14668#supplemental-information.

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
