# Peer review of "CMTM6 is highly expressed in lung adenocarcinoma and can be used as a biomarker of a poor diagnosis"

_PeerJ, doi:10.7717/peerj.14668_

## Round 0.1 · original submission · Major Revisions

All three reviewers have given suggestions. I think the questions and requirements are reasonable. Every question must be taken seriously. We look forward to your revised draft as soon as possible.

Reviewer 1 ·

Basic reporting

The manuscript is well-written, with good references and introduction. However, figures could be greatly improved:
1) Figures 4, 5, ad 8 are illegible and crowded. Authors could consider select one or two panels to focus on and include the rest as supplementary. Also, the authors could summarize the analysis as a bar plot with one-axis reporting -log(FDR).
2) Multiple figure legends should be improved to add in more details. E.g. Figure 7.

Experimental design

1) Authors should demonstrate the specificity of CMTM6 antibody via a knockdown/overexpression experiment.
2) Method section should list number of technical and biological replicates. Also please state number of times an experiment is performed for readers to better judge the reproducibility of the experiment.

Validity of the findings

Please apply multiple test correction whenever appropriate when reporting p-value (E.g. Fig. 5, 6). Especially for GSEA analysis, please report FDR instead of p-value.

Reviewer 2 ·

Basic reporting

In this manuscript, the authors report on the findings that CMTM6 is highly expressed in LUAD and can be used as a poor prognostic biomarker. They have also performed several bioinformatic analyses using the TCGA data and reported some interesting discoveries that could potentially explain the poor prognosis of CMTM6 expression. I find the overall findings in the manuscript clear and important to the field.

Specific comments
1. The authors start in the result section by stating that some tumor samples are positive or negative for CMTM6. However, it is not clear how such a judgment is made. The same applies to the classification of the TCGA datasets. A clear definition of positive/negative, high/low is important for the interpretation of the subsequent analyses.
2. In figure 2, representative IF images for the control cell line should be shown alongside the tumor cell lines.
3. Some references are needed in the main text. For example, on line 219, the authors should cite the TIMER server. The manuscript also contains numerous grammatical errors that need to be corrected.

Experimental design

no comment

Validity of the findings

no comment

Additional comments

no comment

Reviewer 3 ·

Basic reporting

BASIC REPORTING:

The manuscript reports the CMTM6 expression and the lung adenocarcinoma (LUAD) with various staining techniques and RT-PCR method and the results indicate that the CMTM6 are higher in LUAD than the adjacent tissue. The CMTM6 expression with immune cells and immune-checkpoint genes seems to be statistically significant but the correlation coefficient is too low to conclude the relationship between them. The GSEA analysis shows that high expression of CMTM6 regulated the immune signaling system.
Most part of the discussion is mere repetitive of the results. Need to improve the discussion.

Experimental design

Methods:
2.8 Correlation analysis between CMTM6 expression and tumor-infiltrating immune cells in LUAD
Explain how you derived the enrichment score for the immune cells across the expression of CMTM6

Validity of the findings

Result:
3.4 Increased expression of CMTM6 activates immune cell filtrates.
Though the immune cells and the CMTM6 expression is statistically significant, the correlation coefficient (R) indicates weak correlation between the levels of immune cells and CMTM6 expression. Explain why so low correlation coefficient values?

3.5. Correlation analysis of CMTM6 and ICGs
Explain the RNAseq results since you have mentioned the differential expression tool LIMMA. What were the results obtained from LIMMA and how did you select the IGCs based on the RNAseq results?

3.6. CMTM6 expression correlates with immunogenicity and chemosensitivity of LUAD’
The legends in the figure don’t match with the explanation. Check 7B and 7D.

Figures:
Fig 5 (O) Change the Y-axis to indicate the parameter measured.

Tables:
Table 1: Title: Expression of SLC7A5 protein in LUAD and adjacent tissue. The title doesn’t match the table 1 description nor any details of the SLC7A5 protein are mentioned in the results.

Discussion:
The discussion needs to explain and evaluate your findings, showing how it relates to your literature, and making an argument in support of your overall conclusion. It should not be a second results section.

---

## Round 0.2 · accepted · Accept

After the revision, all three reviewers agreed to accept the manuscript. I basically agree with them that this paper has met the requirements of publication and there is no obvious risk of publication.

Reviewer 1 ·

Basic reporting

Authors have well addressed my previous comments.

Experimental design

Authors have well addressed my previous comments.

Validity of the findings

Authors have well addressed my previous comments.

Additional comments

Authors have well addressed my previous comments.

Reviewer 2 ·

Basic reporting

The authors have successfully addressed all my previous comments. Therefore, I recommend publication of this work.

Experimental design

No comment

Validity of the findings

No comment

Additional comments

No comment

Reviewer 3 ·

Basic reporting

The manuscripts reports that CMTM6 is highly expressed in lung adenocarcinoma (LUAD) and can be used as a prognostic marker. The bioinformatic analysis of the TCGA data and the interesting discoveries explain that CMTM6 can be used as a potential biomarker for poor prognosis of LUAD. In my opinion, the author has addressed the comments and corrections and is suitable for publication in the PeerJ journal.

Experimental design

The experimental design makes sense.

Validity of the findings

No comments